# Mining Road Traffic Rules with Signal Temporal Logic and Grammar-Based Genetic Programming

Federico Pigozzi [1], Eric Medvet [1,*] and Laura Nenzi [2,3]

1   Department of Engineering and Architecture, University of Trieste, 34127 Trieste, Italy;
    federico.pigozzi@phd.units.it
2   Department of Mathematics and Geosciences, University of Trieste, 34127 Trieste, Italy; lnenzi@units.it
3   TU Wien Informatics, 1040 Vienna, Austria
*   Correspondence: emedvet@units.it

**Abstract:** Traffic systems, where human and autonomous drivers interact, are a very relevant instance of complex systems and produce behaviors that can be regarded as trajectories over time. Their monitoring can be achieved by means of carefully stated properties describing the expected behavior. Such properties can be expressed using Signal Temporal Logic (STL), a specification language for expressing temporal properties in a formal and human-readable way. However, manually authoring these properties is a hard task, since it requires mastering the language and knowing the system to be monitored. Moreover, in practical cases, the expected behavior is not known, but it has instead to be inferred from a set of trajectories obtained by observing the system. Often, those trajectories come devoid of human-assigned labels that can be used as an indication of compliance with expected behavior. As an alternative to manual authoring, automatic mining of STL specifications from unlabeled trajectories would enable the monitoring of autonomous agents without sacrificing human-readability. In this work, we propose a grammar-based evolutionary computation approach for mining the structure and the parameters of an STL specification from a set of unlabeled trajectories. We experimentally assess our approach on a real-world road traffic dataset consisting of thousands of vehicle trajectories. We show that our approach is effective at mining STL specifications that model the system at hand and are interpretable for humans. To the best of our knowledge, this is the first such study on a set of unlabeled real-world road traffic data. Being able to mine interpretable specifications from this kind of data may improve traffic safety, because mined specifications may be helpful for monitoring traffic and planning safety promotion strategies.

**Keywords:** context-free grammar genetic programming; grammatical evolution; traffic monitoring; formal methods





## 1. Introduction

Autonomous cars are a well-known example of a safety-critical Cyber-Physical System (CPS), meaning that a failure could result in loss of life or in catastrophic consequences for the environment. One of the main challenges for autonomous driving in a real environment consists in ensuring such safety [1]. When many autonomous and human drivers interact in the same environment, achieving safety is even more challenging, due to several factors such as the unpredictability of agents' behavior and scalability.

The explosion of Artificial Intelligence (AI) and Machine Learning permitted the design of very powerful and efficient CPSs. However, it also increased the complexity and opacity of such systems. In real-world systems, where safety plays a crucial role, such opacity and the consequent loss of explainability can be serious issues. Indeed, explainability in AI is itself a prominent goal [2].

In recent years, the formal methods community helped to tackle this problem [3]. Formal methods allow us to describe safe requirements in a formal and rigorous way and to check in an automatic way when such requirements are satisfied. When dealing

with data describing how values vary over time, a possibility is to use temporal logic languages. Temporal logic is a logic that comes with specific temporal operators to describe the temporal evolution of a system. In particular, we consider the Signal Temporal Logic (STL), a logic very suitable to specify properties over real-world trajectories. For example, given two trajectories that describe the velocity of a car and its distance to the closest car over time, with STL, we can formally specify simple properties such as "travel no faster than $30\,\mathrm{km\,h^{-1}}$", as well as more complex properties such as "do not exceed $30\,\mathrm{km\,h^{-1}}$ if, in the last minute, the closest other car has been closer than $30\,\mathrm{m}$". Provided that an STL specification is available, the monitoring of a CPS can be done efficiently using suitable algorithms that can check when the trajectory satisfies the property [4], possibly exploiting distributed processing paradigms [5].

In principle, STL specifications can be designed by the human operator, who should both know the domain, i.e., the involved properties and the expected behavior, and master STL syntax and semantics. In practice, however, designing the right specification that fits the observable system is very hard, because many attributes (and hence many trajectories) should be considered at the same time. A possible solution is to learn the formal specification in an automatic way: given a sufficiently representative set of system trajectories describing the system behavior, some inference technique can be used to synthesize the specification that fits the data. While some approaches for specification inference from data exist (see Section 2 for an overview), the vast majority of them require the user to either (1) specify the structure of the STL specification, while actually inferring only its numerical parameters, or (2) provide both positive and negative example trajectories that are representative of expected and unexpected behaviors. In practical settings, these requirements often become limitations and hamper the applicability of inference techniques.

In this paper, we present a methodology to learn STL specifications from real-world datasets of unlabeled trajectories, i.e., from trajectories for which no human-assigned labels are required that tell if the observations are related to an expected or undesired behavior of the system. Our methodology is based on grammar-based evolutionary optimization, namely Context-Free Grammar Genetic Programming (CFGGP) [6]. We chose this kind of optimization because it is perfectly suited for searching in the space of strings of a language specified by means of grammar: in fact, the STL language can be specified with a grammar (see Section 3). While designing our methodology, we were driven by two goals: generating STL specifications that (1) fit the data and (2) are human-readable. For achieving the first goal, we designed a fitness function, i.e., a measure of the quality of the candidate specification, that measures the degree of (dis)satisfaction of the trajectories with respect to the specification, to be minimized: intuitively, all the trajectories (which, we remark, are unlabeled) should *barely* satisfy or dissatisfy the specification. For achieving the second goal, we structurally limited the complexity of the learned specification, by imposing special constraints within the grammar.

We assess experimentally our methodology on a real-world traffic dataset of 5678 unique vehicles tracked over 19,679 frames with information about velocity and position. After extracting the informative attributes from the dataset, we show how the method can learn STL specifications in an automatic and efficient way. The learned specifications provide relevant information concerning the dataset and are indeed human-interpretable.

The rest of this paper is structured as follows. In the next section, we survey the most significant previous research that is relevant to our study. In Section 3, we briefly introduce the background about STL. In Section 4, we formally define the problem that we aim to solve. In Section 5, we describe our proposed solution, which we experimentally evaluate in Section 6. Finally, we draw the conclusion and we sketch possible future lines of research in Section 7.

## 2. Related Work

There exist several approaches devoted to learning temporal logic specifications, particularly for STL. We partition here them into two categories: template-based and

template-free. While those in the former category rely on a user-provided template formula and focus on estimating parameters for it, the latter also try to learn the formula structure. We fall in the template-free domain.

Template-based methods cast the STL specification mining problem as an optimization problem in terms of the satisfaction degree $\rho$ of an STL formula (see Section 3). Succinctly, we define $\rho$ as a (real-valued) degree of satisfaction of an STL formula with respect to a trajectory. Bartocci et al. [7] adopted an active learning approach, dependent on a probability distribution over $\rho$, to query the next point in the parameters space to be evaluated. Bortolussi and Silvetti [8] extended the work cited previously with a statistical approach that emulates the expected value of this probability distribution using Gaussian Process Regression [9]; optimization of the emulation was then performed via the GP-UCB algorithm [10]. In general, these approaches are labeled as Parametric Signal Temporal Logic (PSTL) [11,12].

Although interesting on its own, the applicability of PSTL is sometimes limited, since specifying templates can be hard to start with. As such, template-free methods also attempt to build an optimal structure for the formula. The vast majority of the works in this regard start from a dataset of labeled trajectories, partitioned into positive and negative examples, and try to learn an STL classifier for the data. As an example, Nenzi et al. [13] proposed ROGE (RObustness GEnetic algorithm), a bi-level optimization procedure, which optimized the structure by a genetic algorithm and the parameters using Bayesian Optimization. To the best of our knowledge, it is the only attempt at using an evolutionary algorithm for solving the template-free problem. Others mined STL structure by exploring a directed acyclic graph [14], using a decision-tree oriented approach [15], or employing enumerative solvers [16].

The achievements of the aforementioned approaches have been remarkable, but none of them addressed the problem of mining STL specifications from unlabeled data. Considering real-world scenarios, it is often the case that we labeled data that are not available, because labeling is costly. Still, those data do bring some information that could be, in principle, condensed in the form of an STL specification. Some special cases of the unlabeled data case have been considered: M. Vazquez-Chanlatte et al. [17] and Mohammadinejad et al. [16] addressed unsupervised clustering of time-series data using PSTL, i.e., they learned just the parameters of template formulas.

To the best of our knowledge, the only work learning both structure and parameters of a formula from unlabeled data is [18], where the authors proposed a heuristic for sequentially building more complex formulas. This seems a very interesting approach, but its applicability to large datasets is unclear: the experimental evaluation of the cited paper is based on a single trajectory and the results concerning the learning of the structure of the formula are not clear. Unfortunately, the code is not available and this hampers reproducibility and comparisons.

A different approach for designing automatic rules for road traffic has been proposed by Medvet et al. [19]. Similar to this study, the authors relied on grammar-based genetic programming and did not use labeled data. Differently from here, they did not use STL as the syntax for the rules and they optimized rules with the goal of maximizing the efficiency and safety of (simulated) road traffic rather than for describing real data of road traffic.

## 3. Background: Signal Temporal Logic

Signal Temporal Logic (STL) is a formal language to specify behaviors of dynamical systems through logic formulas. Let $\mathcal{X}$ be a set of *trajectories* $\boldsymbol{x} : \mathbb{T} \to \mathbb{R}^p$ for every $\boldsymbol{x} \in \mathcal{X}$, with $\mathbb{T} \subseteq \mathbb{R}_{\geq 0}$ a time domain. We say that each trajectory is a $p$-dimensional *signal* of real-valued variables, and we denote by $x_i(t)$ the projection on the $i$-th coordinate of $\boldsymbol{x} = (x_1, \ldots, x_p)$ at time $t \in \mathbb{T}$. We now introduce the *syntax* of STL, the set of rules used for constructing specifications in this language.

**Definition 1** (STL syntax). *We define the syntax of an STL formula $\varphi$ using the following grammar:*

$$\varphi := \top \mid \mu \mid \neg\varphi \mid \varphi_1 \wedge \varphi_2 \mid \varphi_1 S_{[t_1,t_2]} \varphi_2 \mid O_{[t_1,t_2]} \varphi \mid H_{[t_1,t_2]} \varphi,$$

*where $\top$ is the Boolean constant true; $[t_1, t_2]$, with $t_1, t_2 \in \mathbb{T}$, is a time interval such that $t_1 < t_2$; $\mu$ is an atomic proposition of the form $y(t) \sim c$, with $y : \mathbb{R}^p \to \mathbb{R}$ projecting the p-dimensional signal onto a single variable, $\sim \in \{<, >\}$ and $c \in \mathbb{R}^+$ being a threshold (practically inequality over the variable of the signal); $\neg$ and $\wedge$ are the usual Boolean connectives; $S_{[t_1,t_2]}$ is the Since temporal modality; $O_{[t_1,t_2]}$ is the Once temporal modality; and $H_{[t_1,t_2]}$ is the Historically temporal modality.*

The *semantics* of an STL formula $\varphi$ allows us to tell if and to which degree a trajectory $x$ satisfies the formula $\varphi$ at time $t$. We now define two kinds of semantics: with the Boolean semantics, the satisfaction assumes Boolean values (does satisfies, does not satisfy); with the quantitative semantics, the satisfaction assumes real-valued values [20,21].

**Definition 2** (STL Boolean Semantics). *For the Boolean semantics, we write $(x, t) \models \varphi$ if $\varphi$ holds for trajectory $x$ at time $t$. If $\varphi$ is an atomic proposition $\mu$, then $(x, t) \models \varphi$ if and only if $\mu$ is true. The semantics of $\neg\varphi$ and $\varphi_1 \wedge \varphi_2$ is trivial and the semantics of Since is defined as follows:*

$$(x, t) \models \varphi_1 S_{[t_1,t_2]} \varphi_2 \Leftrightarrow \exists t' < t : ((x, t') \models \varphi_2 \wedge \forall t'' \in [t', t[ : (x, t'') \models \varphi_1),$$

*with $t' \in [t_1, t_2]$. In other words, we say that $\varphi_1 S_{[t_1,t_2]} \varphi_2$ is satisfied at time $t$ if $\varphi_2$ occurs at some point in $[t_1, t_2]$ and $\varphi_1$ holds continuously since then. The other temporal operators are defined based on Since: Once as $O_{[t_1,t_2]}\varphi = \top S_{[t_1,t_2]}\varphi$ and Historically as $H_{[t_1,t_2]}\varphi = \neg O_{[t_1,t_2]}\neg\varphi$.*

**Definition 3** (STL Quantitative Semantics). *The quantitative satisfaction function $\rho$ returns a value $\rho(\varphi, x, t) \in \mathbb{R} \cup \{-\infty, +\infty\}$ quantifying the robustness degree of the formula $\varphi$ by the trajectory $x$ at time $t$. It is defined recursively as follows:*

$$\rho(\top, x, t) = +\infty$$
$$\rho(\mu, x, t) = y(x(t)) \quad where \ \mu \equiv y(x(t)) \geq 0$$
$$\rho(\neg\varphi, x, t) = -\rho(\varphi, x, t)$$
$$\rho(\varphi_1 \wedge \varphi_2, x, t) = \min(\rho(\varphi_1, x, t), \rho(\varphi_2, x, t))$$
$$\rho(\varphi_1 S_{[t_1,t_2]} \varphi_2, x, t) = \sup_{t' \in [t-t_1, t-t_2]} \left( \min\left( \rho(\varphi_2, x, t'), \inf_{t'' \in [t', t[} \rho(\varphi_1, x, t'') \right) \right).$$

*Similar to the Boolean semantics, the Once and Historically temporal operators are defined based on Since.*

The sign of $\rho(\varphi, x, t)$ provides the link with the standard Boolean semantics. It holds that $\rho(\varphi, x, t) > 0$ if and only if $x \models \varphi$, while $\rho(\varphi, x, t) < 0$ if and only if $x \not\models \varphi$. The case $\rho(\varphi, x, t) = 0$, instead, is a borderline case, and the truth of $\varphi$ cannot be assessed from the robustness degree alone.

In practical settings, systems are monitored for a given, finite amount of time, and, as a result, trajectories have a limited time span. We define as $|x|$ the length of a trajectory, i.e., its number of samples. For the sake of this study, monitoring an STL formula $\varphi$ over a trajectory $x$ is constrained to $[0, |x| - 1] \subset \mathbb{T}$, a subset of the time domain.

When talking about temporal formulas, the *necessary length* concept is of importance [20]. The necessary length of a formula $\varphi$ (let it be $\|\varphi\|$) is defined recursively as:

$$\|\mu\| = 0$$
$$\|\neg\varphi\| = \|\varphi\|$$
$$\|\varphi_1 \wedge \varphi_2\| = \max(\|\varphi_1\|, \|\varphi_2\|)$$
$$\|\varphi_1 S_{[t_1,t_2]} \varphi_2\| = \max(\|\varphi_1\|, \|\varphi_2\|) + t_2$$

Intuitively, the necessary length is the shortest trajectory length such that $(x, t) \models \varphi$ is well-defined. For example, the formula $\varphi_1 S_{[0,10]} \varphi_2$ cannot be evaluated on trajectories shorter than 10 (assuming that $\|\varphi_1\| > 10$ and $\|\varphi_2\| > 10$), since this would imply looking at a future that is not part of the trajectory.

## 4. Problem Statement

We consider systems described by real-valued attributes and a set of trajectories that describe the way these attributes vary over time. We aim to mine specifications that (1) describe such trajectories (2) in a way that the specifications are readable and interpretable for a human.

Formally, let $X = \{x_1, \ldots, x_n\}$ be a set of trajectories gathered from a system described by attributes $A = \{a_1, a_2, \ldots, a_{|A|}\}$. Let $\Phi$ be the space of all possible STL formulas defined over $A$. We aim at finding a $\varphi^\star \in \Phi$ that is both human-readable and describes $X$. More specifically, $\varphi^\star$ should be *tight* with respect to $X$, i.e., the robustness value $|\rho(\varphi^\star, x_i, t)|$ should be as much as possible close to zero for all $x_i \in X$ and $t$. From another point of view, small perturbations on tight formulas $\varphi^\star$ should result in an overall increase in robustness for some trajectories of $X$; i.e., some trajectories could be described by a tighter formula $\varphi' \neq \varphi^\star$.

Tightness is of fundamental importance when no labels are provided; in fact, the satisfaction of a robustness metric can be trivially maximized by "pushing" the parameters towards the boundaries of the parameter space, resulting in rules that are of no relevance. For example, if $a < 0$ is satisfied by all trajectories, then $a < 1000$ will be satisfied as well and have a much greater degree of robustness.

Concerning human-interpretability, a formal, widely accepted definition for STL formulas does not exist. Indeed, the lack of such a definition holds for many other kinds of Machine Learning modules. Moreover, it is acknowledged that human-interpretability is not, itself, universal: different subjects might perceive the same model as differently interpretable [22]. In this work, we circumvent the problem of defining interpretability with a practical approach: we force our method to search in a subset of $\Phi$ containing structurally simple formulas, where both the overall size of the formula and the maximum degree of nesting of temporal operators are limited. The rationale of the latter constraint is in the fact that the interpretability of symbolic models is affected differently by different kinds of components [23].

We remark that our problem is more complex than the mere specification mining of a template formula where the structure of the formula is already fixed (see Section 2), which searches only in the parameter space of the formula. In the easier case, the user has to provide a template formula $\hat{\varphi}$, and the problem is a numerical optimization problem; i.e., the solution has to be found in $\mathbb{R}^m$, where $m$ is the number of numerical parameters in $\hat{\varphi}$. In our case, no burden is on the user, but the system is required to search in $\Phi$ rather than $\mathbb{R}^m$. In Section 5, we detail the methodology we employed to address this problem statement.

## 5. Methodology

Since the STL syntax can be defined by means of context-free grammar (CFG), we rely on CFGGP [6], a grammar-based version of GP [24]. In CFGGP, candidate solutions are represented as derivation trees of a grammar $\mathcal{G} = (N, T, s_0, R)$, where $N$ is the set of non-terminal symbols, $T$ is the set of terminal symbols (with $T \cap N = \varnothing$), $s_0 \in N$ is the starting symbol, and $R$ is the set of derivation rules. Each derivation rule describes how a non-terminal symbol may be replaced by a sequence of symbols, either terminal or non-terminal. A *derivation tree* is a tree where nodes are symbols of the grammar: leaf nodes are terminal symbols, and non-leaf nodes are non-terminal symbols. The children of each node match one of the derivation rules for the corresponding non-terminal symbol. The string of the language defined by the grammar corresponding to a given derivation tree is the sequence of the leaves of the tree. Note that a derivation tree is not an abstract syntax tree: the former is in general deeper than the latter, for the same formula. Figure 1 shows an example of a derivation tree for the CFG of Figure 2.

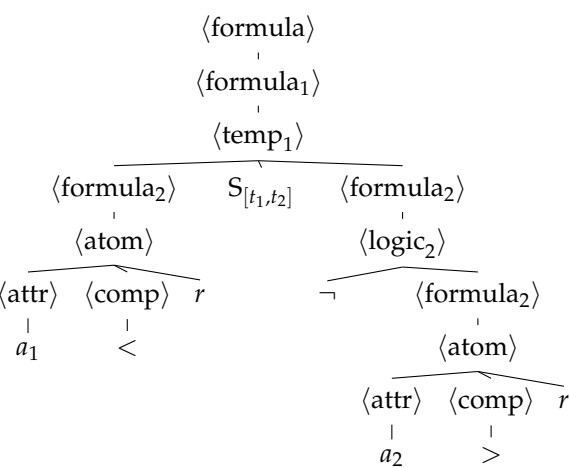

**Figure 1.** A derivation tree of the grammar of Figure 2 for the formula $(a_1 < r)S_{[t_1,t_2]}\neg(a_2 > r)$.

We use an improved version of CFGGP that promotes diversity in the population. The lack of diversity in the population may result in premature convergence towards a local optimum [25], in particular when the search space is discrete, as in CFGGP [26]. In this work, we promote diversity by simply enforcing the re-application of the genetic operator whenever a generated individual is already part of the population.

*5.1. Evolutionary Algorithm*

Given a CFG $\mathcal{G}$ and a fitness function $f : L(\mathcal{G}) \to \mathbb{R}$, CFGGP works as shown in Algorithm 1. After the initialization of the population $P$, CFGGP repeats $n_{\text{gen}}$ times the following three steps.

1. It builds the offspring population $P'$, with $|P'| = n_{\text{pop}}$, by iteratively selecting one (mutation, with $1 - p_{\text{xover}}$ probability) or two (crossover, with $p_{\text{xover}}$ probability) parents chosen with tournament selection of size $n_{\text{tour}}$ and then applying the genetic operator. If the resulting solution $\varphi_c$ is already part of the offspring $P'$ or parent population $P$, a new solution is generated, and the process is repeated for a maximum number of $n_{\text{atts}}$ attempts; otherwise $s_c$ is added to $P'$ and its fitness $f(\varphi)$ is computed.
2. It merges the parent and offspring populations $P'$ and $P$.
3. It shrinks the resulting new population $P$, until its size is $n_{\text{pop}}$, by iteratively removing the worst solution.

The initial population is built with the ramped half-and-half method [27]. Let a range $\{d_{\min}, \ldots, d_{\max}\}$ for the depth of the derivation trees be given and let $n_{\text{pop}}$ be the number of trees to be generated. For each $d$ in the range, we build $k$ random approximately full derivation trees (i.e., where each leaf node is at depth $d$) and $k$ random trees with the deepest leaf at depth $d$, with $k = \frac{n_{\text{pop}}}{2(d_{\max}-d_{\min}+1)}$. We write "approximately" because it is not possible, in general, to build a derivation tree of a grammar $\mathcal{G}$ where each leaf is exactly at depth $d$. This procedure ensures that the size, and hence, the complexity of the generated formulas is evenly distributed in a predefined range.

The genetic operators are defined over the space of derivation trees of the grammar $\mathcal{G}$. We used the standard CFGGP mutation and crossover. The former "replaces" a random subtree of the derivation tree with a randomly generated subtree that is appropriate according to $\mathcal{G}$. The crossover "replaces" a random subtree of one parent with an appropriate random subtree of the other parent. In both cases, it is ensured that the resulting derivation tree is at most $d_{\max}$ deep.

---

**Algorithm 1:** The EA for the optimization.

**function** evolve():
    $P \leftarrow$ initialize$(\mathcal{G}, n_{\text{pop}})$
    **foreach** $i \in \{1, \dots, n_{\text{gen}}\}$ **do**
        $P' \leftarrow \varnothing$
        **while** $|P'| \leq n_{\text{pop}}$ **do**
            $i \leftarrow 0$
            **repeat**
                **if** $\sim U(0,1) \leq p_{\text{xover}}$ **then**
                    $(\varphi_{p,1}, f_{p,1}) \leftarrow$ select$(P)$
                    $(\varphi_{p,2}, f_{p,2}) \leftarrow$ select$(P)$
                    $\varphi_c \leftarrow$ crossover$(\varphi_{p,1}, \varphi_{p,2}; \mathcal{G})$
                **else**
                    $(\varphi_p, f_p) \leftarrow$ select$(P)$
                    $\varphi_c \leftarrow$ mutate$(s_p; \mathcal{G})$
                **end**
                $i \leftarrow i + 1$
            **until** $(\varphi_c \notin P \cup P') \wedge (i \leq n_{\text{atts}})$
            $P' \leftarrow P' \cup \{(\varphi_c, f(\varphi_c)\}$
        **end**
        $P \leftarrow P \cup P'$
        **while** $|P| \geq n_{\text{pop}}$ **do**
            $P \leftarrow P \setminus \{$worst$(P)\}$
        **end**
    **end**
    **return** best$(P)$
**end**

---

### 5.2. Fitness Function

To achieve the goals of Section 4, we use as fitness function:

$$f(\varphi) = \frac{1}{|\mathcal{X}|} \sum_{x \in \mathcal{X}} \left| \rho(\varphi, x, t_f) \right|,$$

which computes the average absolute quantitative robustness of an individual $\varphi$ over the dataset $\mathcal{X}$. Minimizing this quantity is consistent with the notion of achieving a tight evaluation with respect to the trajectories in $\mathcal{X}$. Tightness is of importance when no labels are provided; in fact, the satisfaction of a robustness metric can be trivially maximized by "pushing" the parameters towards the boundaries of the parameter space, resulting in rules that are of no relevance. As a matter of example, if $a < 0$ is satisfied by all trajectories, then $a < 1000$ will be satisfied as well and have a much greater degree of robustness. By minimizing the sum of the absolute values, $f$ achieves a tight evaluation as it rewards individuals $\varphi$ having robustness values as close as possible to zero. Finally, we divide by the total number of trajectories so that, for normalized data, $f \in [0, 1]$, with 0 corresponding to a formula that perfectly fits all of trajectories, and 1 corresponding to a formula that does not satisfy all of the trajectories in the worst possible way.

### 5.3. Grammar for STL Formula Structures

We need to define a grammar $\mathcal{G}$ for the language of STL formulas. $\mathcal{G}$ must be customizable for the considered problem, i.e., for its attributes $A$, and must allow generating formulas along with appropriate values as numerical parameters.

In order to favor the building of human-readable formulas, we build the grammar to explicitly limit the depth of nesting of temporal operators. We remark that the overall size of STL formulas is limited by the value of $d_{\text{max}}$ used by CFGGP (in the operators and in the

initialization of the population). However, we believe that posing a further limit on the composition of the temporal operators may make the STL formulas more readable, and not only just smaller. Our belief is corroborated by the findings of [23] for mathematical expressions: some operators, such as log and sin, make the expressions less interpretable than others, e.g., $+$ and $\div$.

Figure 2 shows the grammar for STL formula structures with limited nesting of the temporal operators. The figure adopts the common Backus–Naur form: the non-terminal symbols are enclosed in angle brackets, whereas the terminal symbols are shown as literals ($\neg$, $\wedge$, ..., $r$, $a_1$, $a_2$, ..., $<$, $>$, $0$, ..., $9$); for each non-terminal, derivation rules are separated by $|$; the starting symbol is the topmost non-terminal, i.e., $s_0 = \langle \text{formula} \rangle$. The terminals $a_1$, $a_2$, ..., derived from the non-terminal $\langle \text{attr} \rangle$, represent the attributes of the problem at hand: in this way, the grammar is tailored to a specific problem. For brevity, we express some of the non-terminals using a parameter $i$ that represents the maximum nesting. The only derivation rule that increases $i$ is the one for $\langle \text{temp}_i \rangle$, which represents (partial) formulas with temporal operators. The limit to nesting is enforced by the parametric definition of the derivation rule of $\langle \text{formula}_i \rangle$, that does not expand to $\langle \text{temp}_i \rangle$ if $i \geq i_{\max}$. In this study, we set the maximum nesting to $i_{\max} = 3$. This means that CFGGP operates on a grammar $\mathcal{G}$ that is the realization of the grammar of Figure 2 with $i_{\max} = 3$ and a given set of attributes $A$.

$$\langle \text{formula} \rangle ::= \langle \text{formula}_1 \rangle$$

$$\langle \text{formula}_i \rangle ::= \begin{cases} \langle \text{atom} \rangle \mid \langle \text{logic}_i \rangle \mid \langle \text{temp}_1 \rangle & \text{if } i < i_{\max} \\ \langle \text{atom} \rangle \mid \langle \text{logic}_i \rangle & \text{otherwise} \end{cases}$$

$$\langle \text{logic}_i \rangle ::= \neg \langle \text{formula}_i \rangle \mid \langle \text{formula}_i \rangle \wedge \langle \text{formula}_i \rangle$$

$$\langle \text{temp}_i \rangle ::= \langle \text{formula}_{i+1} \rangle \text{S} \langle \text{interval} \rangle \langle \text{formula}_{i+1} \rangle \mid$$
$$\text{H} \langle \text{interval} \rangle \langle \text{formula}_{i+1} \rangle \mid \text{O} \langle \text{interval} \rangle \langle \text{formula}_{i+1} \rangle$$

$$\langle \text{interval} \rangle ::= [\langle \text{num} \rangle, \langle \text{num} \rangle]$$

$$\langle \text{atom} \rangle ::= \langle \text{attr} \rangle \langle \text{comp} \rangle \langle \text{num} \rangle$$

$$\langle \text{attr} \rangle ::= a_1 \mid a_2 \mid \ldots \mid a_{|A|}$$

$$\langle \text{comp} \rangle ::= \; < \; \mid \; >$$

$$\langle \text{num} \rangle ::= \langle \text{digit} \rangle \langle \text{digit} \rangle$$

$$\langle \text{digit} \rangle ::= 0 \mid 1 \mid 2 \mid 3 \mid 4 \mid 5 \mid 6 \mid 7 \mid 8 \mid 9$$

**Figure 2.** The CFG for describing STL formula structures. Non-terminal symbols are enclosed in angle brackets: the topmost non-terminal symbol, $\langle \text{formula} \rangle$, is the starting symbol $s_0$ of the grammar. The derivation rules for the symbols $\langle \text{formula}_i \rangle$, $\langle \text{logic}_i \rangle$, $\langle \text{temp}_i \rangle$ are parametric on $i$, which represents the nesting level. The derivation rule for $\langle \text{attr} \rangle$ is the one that makes the grammar tailored to a given system with attributes $A = \{a_1, a_2, \ldots, a_{|A|}\}$.

When mapping a derivation tree into the corresponding STL formula, we apply the following adjustments concerning numerical parameters. To map a non-terminal interval symbol $\langle \text{interval} \rangle$ into the corresponding time interval, we map the first $\langle \text{num} \rangle$ into the corresponding integer in $\{0, \ldots, 99\}$ and set it to be the start of the interval. We then map the second $\langle \text{num} \rangle$ into the corresponding integer in $\{0, \ldots, 99\}$ and add it to the start to obtain the end of the interval. As such, we avoid any issue arising from intervals having a start greater or equal to the end. Moreover, we remark that, since there can be at most two nested temporal operators, the maximum necessary length of the formula will be 198, corresponding to the necessary length of a formula with two nested temporal operators with intervals $[0, 99]$. When mapping a non-terminal $\langle \text{atom} \rangle$, we divide the product of its $\langle \text{num} \rangle$ (the numeric constant in the atomic proposition) by 100. As a result, numeric constants lie in $[0, 1]$, and, for normalized data, we can express all possible thresholds.

## 6. Experimental Evaluation

Considering the goals described in Section 4, we aim at answering the following research questions:

RQ1　Can we mine specifications that describe the input unlabeled trajectories?
RQ2　Are the mined specifications readable and interpretable for a human?

We consider to formula describe the dataset well if it tightly fits the pool of trajectories. To this end, we verify whether the fitness $f$ of the learned formula is as close as possible to 0.0. We say that a formula is readable and interpretable for a human if it is parsimonious; to this end, we verify whether the size of a formula (number of nodes of the derivation tree), $|\varphi|$, is reasonable. Moreover, we also verify whether a formula is easily understandable by a human by manually inspecting and reporting it.

To answer the research questions according to these definitions, we ran an experimental campaign on real-world data of road traffic. We performed 10 evolutionary runs with different random seeds. We used the same parameter values for all the runs and set $n_{\text{pop}} = 500$, $n_{\text{gen}} = 50$, $n_{\text{tour}} = 5$, $p_{\text{xover}} = 0.8$, $n_{\text{atts}} = 100$, $d_{\text{min}} = 1$, and $d_{\text{max}} = 12$.

We executed the experiments on an HPC cluster with nodes equipped with $2 \times 18$ cores based on 2.30 GHz Intel Xeon E5-2697 v4 (Broadwell) and with 128 GB RAM. Fitness evaluations were parallelized across cores; evolutionary runs were parallelized across nodes. We implemented the software for the experiments in the Java programming language and made it publicly available at https://github.com/pigozzif/STLRulesEvolutionaryInferenceNoClass (accessed on 1 November 2021). The project employs the monitoring tool MoonLight (https://github.com/MoonLightSuite/MoonLight, accessed on 1 November 2021) [28] and the evolutionary framework JGEA (https://github.com/ericmedvet/jgea, accessed on 1 November 2021).

### 6.1. Data

The dataset used in this study [29] consists of the trajectories of all the vehicles crossing the eastbound I-80 Freeway in Emeryville, California (USA). Measurements date back to 13 April 2005 and were taken on a tract of approximately 1640 ft, comprising six lanes and an on-ramp. The full dataset is partitioned into three 15 min sequences: 4:00 p.m. to 4:15 p.m., 5:00 p.m. to 5:15 p.m., and 5:15 pm to 5:30 p.m. Intuitively, they correspond to different traffic patterns, from the build-up of congestion to the peak period.

The dataset contains a total of 5678 unique vehicles tracked over 19,679 frames. For each vehicle and each frame, it contains the position of the vehicle (lateral and longitudinal offsets with respect to a reference position), its velocity, its size (width and length), and a lane identifier. All these attributes have been extracted by the creators of the dataset by means of image processing and computer vision techniques: we refer the reader to [29] for more details. Figure 3 provides a graphical representation of the information contained in a frame of the dataset.

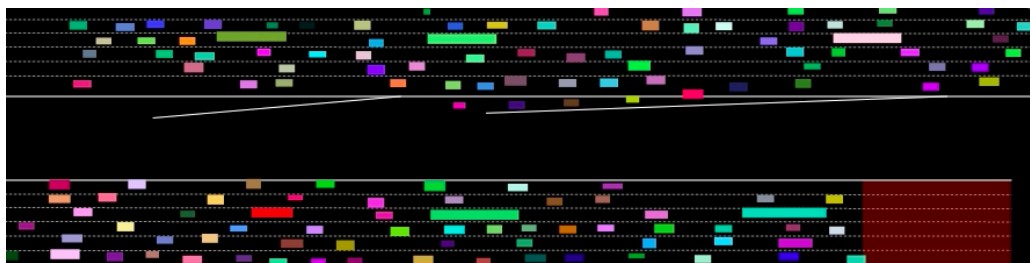

**Figure 3.** Sample frame reproducing the traffic of the dataset [29]. Each colored box represents a car. Dotted lines are lane separators, while solid lines are guardrails. The two segments projecting out from the first level of the road are the boundaries of the on-ramp. The second level of road is the continuation of the top one, while the red shaded rectangle is the range for the trajectory endpoints.

*6.2. Data Processing*

The aim of the data processing step is to extract useful attributes. Chiefly, we want attributes that (1) are more meaningful to the other road settings than the one considered in this study and (2) effectively describe the phenomenon at hand, i.e., capture measurements that are relevant for monitoring road traffic. The first point discards attributes such lane identifier and position with respect to the reference point since they would be of no interest for roads with a different topology and number of lanes. At the same time, we want attributes that are relative to the vehicle and not to the setting. For example, positions and coordinates should be relative, not absolute. As a result, formulas are more readable as they employ attributes that are immediately comprehensible. In the following, we detail how we extracted additional attributes from the ones reported in the original dataset.

A set of very relevant attributes is the set of distances from the nearest neighbors of each vehicle. Intuitively, such a set of attributes is relevant for drivers and allows formulas to clearly and synthetically be expressed such as " keep a safety distance of at least $10\,\text{m}$ from the closest front vehicle". To formalize this, we partition the space surrounding each vehicle as shown in Figure 4, and we find the closest vehicle in each of the eight regions. We thus consider eight new attributes, namely E, SE, S, SW, W, NW, N, and NE.

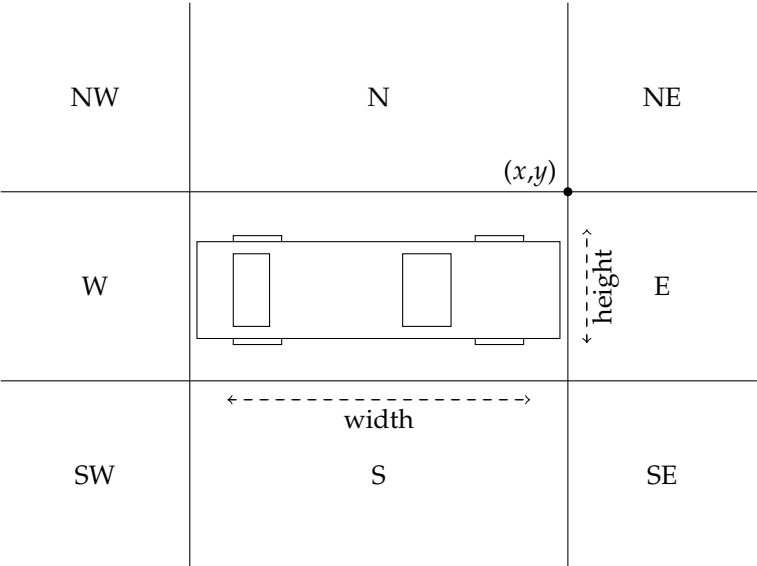

**Figure 4.** A car and its eight neighboring regions. Regions are labeled using cardinal directions. Boundaries can be swiftly computed starting from the $x, y$ positions of the front-left corner of the car, using car width and car height (provided in the dataset).

To efficiently compute them, we used a kd-tree [30], a spatial data structure. Given a query point and a set of $k$-dimensional points, it allows for logarithmic-time (in the number of points) look-up of nearest neighbors, improving over the quadratic time (in the number of points) of a brute-force search. For us, points are cars and are bi-dimensional (lateral and longitudinal offsets with respect to the reference position). Given that construction and update take both linear time (in the number of points) for a kd tree, we built a separate tree for every frame instead of updating all cars' positions.

Finally, after having found the nearest neighbors for every car and for every frame, we trimmed neighbors that might not be sensed in a real-world setting. In particular, we set to $-\infty$ the distances above $485\,\text{ft}$. Moreover, we limit the neighbor search to the current and adjacent lanes of a vehicle, as we assume that drivers do not care about vehicles in far lanes.

To summarize, we built a dataset with velocity (vel) and eight distances (E, SE, S, SW, W, NW, N, NE) as attributes, which together form the set of attributes $A$ mentioned in Section 4. On the other hand, width and height were discarded.

We do not consider trajectories being longer than $t_f = 20\,\mathrm{s}$, since that could result in specifications requiring agents (both human and automated) to consider an unpractically long behavior history. To accomplish this, we partitioned the trajectories as follows. Let $|x|$ be the number of samples for trajectory $x$. For each trajectory $x \in \mathcal{X}$, we split it into a new set of trajectories by sliding a window of size $n$ over it, with $\frac{n}{2}$ overlapping, obtaining approximately $\frac{2|x|}{n}$ new trajectories. The resulting pool of trajectories shares the same length $n$, which must be chosen to reflect a sensible interval of monitoring for an autonomous agent. Thus, we set $n = 200$, corresponding to 20 s for this dataset. We remark that, by fixing $n = 200$, all the new trajectories will have a maximum size (number of samples) of 200; given that, with our grammar, the maximum necessary length of the formula is 198 (see Section 5.3), no issue regarding the necessary length of formulas arises.

Finally, as a consequence of our grammar shown in Figure 2, and to avoid favoring attributes with shorter ranges, we normalize attributes to $[0, 1]$ by subtracting the minimum and dividing by the range.

### 6.3. Results

#### 6.3.1. RQ1: Solutions that are Effective

We ran an experimental campaign with the aforementioned parameters. To evaluate whether our approach is effective, we (1) verify whether it evolved tight formulas, and (2) compare the fitness $f$ of the best individuals with that of randomly initialized formulas. For the former, Figure 5 plots the histogram for the distribution of $\rho(\varphi, x, t)$ for the best individuals $\varphi$ found in each run (i.e., the individual having the lowest fitness $f$ at the last generation) and for all the car trajectories $x$. For the latter, Figure 6 reports median $\pm$ standard deviation fitness $f$ over the course of evolution for the best individuals found in each run. Table 1 summarizes (with median $\pm$ standard deviation) the results in terms of fitness $f$ and $|\varphi|$ for the best individual found in each run, as well as evolution time (in seconds).

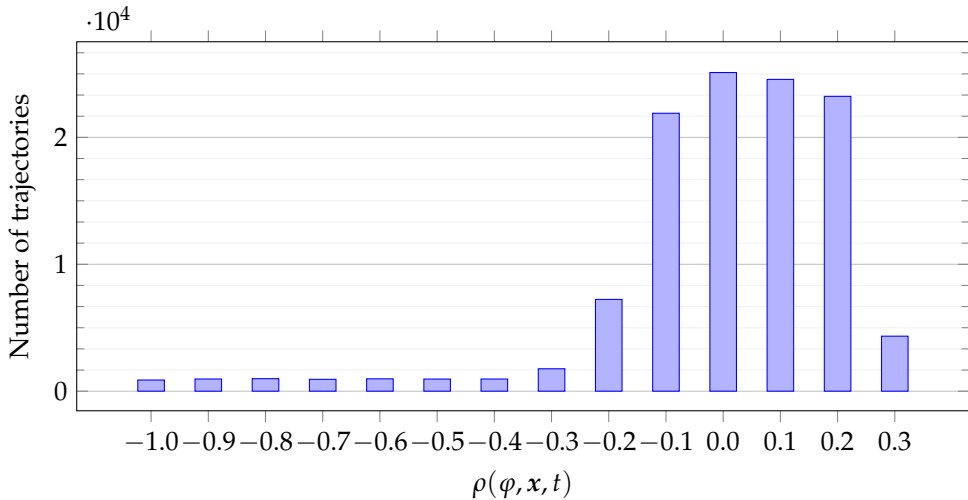

**Figure 5.** Distribution of the robustness $\rho(\varphi, x, t)$, computed for all the I-80 trajectories $x$, for the best individual $\varphi$ found in each run.

From Figure 5, two observations can be made. First, robustness values cluster around 0.0. Second, robustness values cluster symmetrically. Table 1 corroborates these observations. Recall from Section 4 that our goal is to mine formulas that are tight since tightness is of fundamental importance when no labels are provided. Formulas are tight when robustness values are as close to zero as possible, in particular $|\rho(\varphi, x_i, t_f)| < \epsilon, \forall i$, with $\epsilon$ a small quantity, which depends on the system at hand. In other words, formulas must fit the pool of trajectories at hand and in such a way that small perturbations make the robustness value greater than $\epsilon$ on some trajectory. By looking at Figure 5, we remark that

the mined formulas are indeed tight. Let a value $\epsilon$ be fixed. The best individuals produce robustness values that fit into a segment centered in 0.0 and of length $2\epsilon$; that is, they lie in $[-\epsilon, +\epsilon]$. Choose an adequate value for $\epsilon$, e.g., $\epsilon = 0.25$. From Figure 5, we notice that the vast majority of values lie in $[-0.25, 0.25]$. Recalling our definition of tightness of a formula from Section 4, these results confirm that the best individuals indeed are tight formulas.

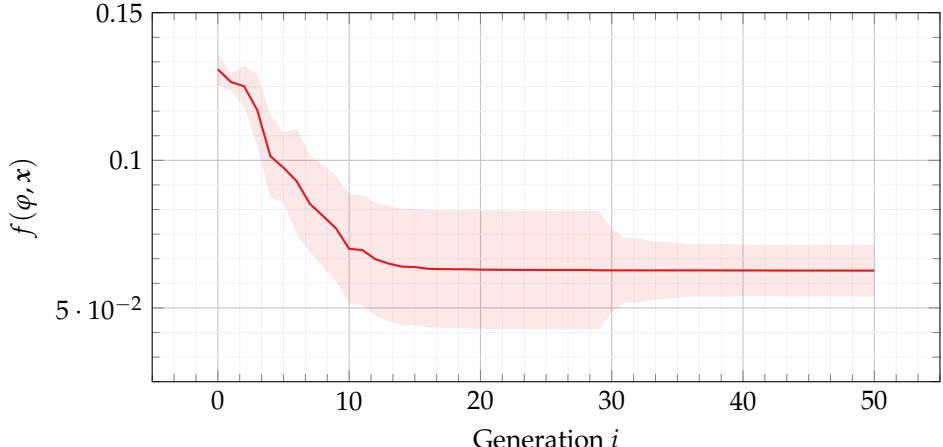

**Figure 6.** Median $\pm$ standard deviation (across the 10 evolutionary runs) over the evolution.

**Table 1.** Fitness $f$, solution (derivation tree) size $|\varphi|$ for the best individuals found in each run, and evolution time in seconds. Reported as median $\pm$ standard deviation.

| $f$ | $|\varphi|$ | Time [s] |
|---|---|---|
| $0.063 \pm 0.009$ | $52.5 \pm 6.5$ | $3876.6 \pm 426.4$ |

As a further confirmation that our methodology actually learns tight formulas, we also show that learned formulas are significantly better than random formulas. As can be seen from Figure 6, fitness $f$ progresses over the course of evolution and settles into a (local) optimum. Considering that the initial population is composed entirely of randomly initialized formulas (see Section 5.1), this observation points to the fact that, in terms of fitness $f$ (and, thus, of tightness), the best individuals are clearly better than random formulas.

To comment, our methodology succeeds in mining specifications that are effective and tight with respect to the pool of trajectories.

### 6.3.2. RQ2: Specifications that are Readable and Interpretable for a Human

As far as readability is concerned, we found the mined specifications to be readable and interpretable by a human. To provide deeper insights, Figure 7 plots the histogram for the frequency distribution of the operators (see Section 5.3) and the attributes (see Section 6.2) in our grammar.

By manually inspecting the best individuals and visualizing the aggregate frequencies with Figure 7, we found that all the attributes are present, with a slight overabundance of E and W that are, respectively, the distance from the front neighbor and the distance from the rear neighbor. Moreover, evolution prefers the $\wedge$ and $\neg$ operators over the temporal operators (of which there is approximately one per formula). This finding is in line with our expectations since temporal operators are likely to be the least interpretable for a human (as confirmed for mathematical expressions in [23]).

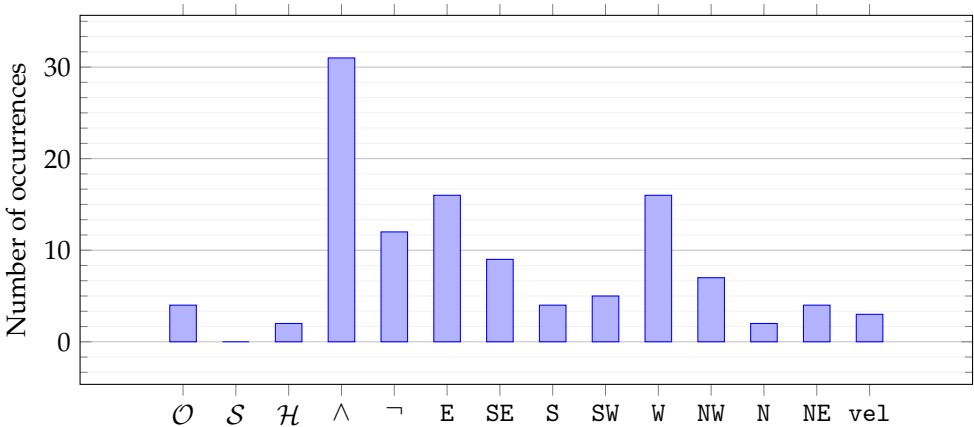

**Figure 7.** Number of occurrences of operators and attributes for the best individual of each evolutionary run.

In the following, we transcribe some instances of best individuals. For ease of understanding, the numeric constants that appear in the formulas below have been denormalized by multiplying by the attribute range and summing the attribute minimum value. For example,

$$(O_{[15,78]}32.85 < \text{SE} < 182.38) \wedge (\neg((\text{SE} < 133.13) \wedge (\text{SW} > 177.77)))$$

dictates a reasonable distance from the southeast neighbor (i.e., between 32.85 ft and 182.38 ft) at least once in the $[15, 78]$ past temporal window; at the same time, such a distance must not be lower than 133.13 ft and the distance from the southwest neighbor higher than 177.77 ft at the same time, showing that distances from neighbors must be, in general, balanced. Similarly, this other example:

$$(O_{[1,87]}\text{SE} > 41.23) \wedge (\neg((\text{E} < 158.36) \wedge (\text{N} > 42.48)))$$

dictates to stay farther than 41.23 ft from the southeast neighbor, while staying not closer than 42.48 ft to the north one and not farther than 158.36 ft from the east one. An example with Historically is:

$$(\neg(\text{W} < 129.48)) \wedge (H_{[41,81]}\text{N} < 71.41) \wedge (\text{SE} < 88.87)$$

which tells it to stay farther than 129.48 ft from the west neighbor, while keeping closer than 88.87 ft from the southeast neighbor and continuously over $[41, 81]$ closer than 71.41 ft from the northern neighbor. An example without temporal operators is:

$$(\text{S} < 69.59) \wedge (\text{E} > 25.12) \wedge (\neg(\text{NW} < 104.98)) \wedge (\text{vel} < 37.93)$$

which tells it to drive slower than 37.93 ft/s while keeping not too far away from the southern neighbor and reasonably far from the east and northwest neighbors. Finally,

$$(35.03 < \text{NW} < 136.51) \wedge (\text{SE} > 82.69) \wedge (\neg(\text{SW} < 286.96))$$

compactly dictates to stay at a distance that is neither too far away nor too close to the northwest, southeast, and southwest neighbors.

From the examples above, we draw one very important conclusion. Recalling Section 6.1 and Figure 3, the dataset used here was collected on a very trafficked highway. The mined specifications follow a common pattern of the ruling:

(i)     to poise the distances from the neighbors, and
(ii)    to drive neither too fast nor too slow.

The former point is crucial, as keeping too far from one neighbor implies coming very close to the neighbors on the opposite side.

To summarize, the mined specifications closely mimic what a car stuck in dense traffic would do and point to the effectiveness of our approach. We believe the reason for such readability to be the parsimony of the mined STL formulas. Intuitively, parsimony is directly linked to interpretability. In fact, as reported in Table 1, median solution size $|\varphi|$ is 52.5, which is a reasonable size for an STL formula.

## 7. Conclusions and Future Work

We have considered the case of monitoring and describing the behavior of traffic systems by means of Signal Temporal Logic (STL) formulas. Authoring these formulas is a hard task due to the necessity of knowing the system at hand and mastering language syntax. Automatically learning STL formulas would allow for the real-time monitoring of traffic systems with the result of improving safety and providing an explanation for the behaviors of autonomous agents. We endeavor to do so with the goal of learning formulas that describe the system at hand and are interpretable for a human.

We proposed a methodology to learn STL formulas for real-world traffic trajectories; the trajectories are unlabeled, in the sense that there are no human-assigned labels discriminating between positive and negative behaviors. Since the STL language can be specified by means of grammar, we use a grammar-based evolutionary optimization algorithm to evolve STL formulas. We evaluate formulas against a fitness function that rewards those that tightly fit the pool of trajectories at hand.

With an experimental campaign, we showed that our approach (1) learns formulas that tightly fit the pool of trajectories and (2) appears interpretable to a human due to its parsimony. We believe that, by applying our approach for inferring formulas describing road traffic in different conditions (e.g., different countries), one could systematically compare alternatives using formulas instead of raw data.

In the future, we will extend our approach to supervised binary classification scenarios, i.e., scenarios in which trajectories come accompanied by human-assigned labels discriminating between positive and negative behaviors. We will also consider anomaly detection, in which only a subset of the positive trajectories is labeled and we want an STL classifier to correctly detect negative trajectories.

**Author Contributions:** Conceptualization, F.P., E.M. and L.N.; methodology, F.P., E.M. and L.N.; software, F.P., E.M. and L.N.; validation, F.P.; formal analysis, F.P., E.M. and L.N.; investigation, F.P., E.M. and L.N.; resources, not applicable; data curation, F.P.; writing—original draft preparation, F.P., E.M. and L.N.; writing—review and editing, F.P., E.M. and L.N.; visualization, F.P.; supervision, E.M. and L.N.; project administration, E.M.; funding acquisition, E.M. and L.N. All authors have read and agreed to the published version of the manuscript.

**Funding:** The experimental evaluation of this work has been done on CINECA HPC cluster within the CINECA–University of Trieste agreement. F.P. was partially supported by a Google Faculty Research Award granted to E.M. L.N. was partially supported by the Austrian FWF projects ZK-35 and by the Italian PRIN project "SEDUCE" n. 2017TWRCNB.

**Institutional Review Board Statement:** Not applicable.

**Informed Consent Statement:** Not applicable.

**Data Availability Statement:** The software for the experiments reported in this paper is publicly available at https://github.com/pigozzif/STLRulesEvolutionaryInferenceNoClass (accessed on 1 November 2021). The data is publicly available and provided by [29].

**Conflicts of Interest:** The authors declare no conflict of interest.

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
