# Peer review of "Mining Road Traffic Rules with Signal Temporal Logic and Grammar-Based Genetic Programming"

_applsci, doi:10.3390/app112210573_

Round 1

Reviewer 1 Report

The paper is well-structured. Figures 4,5, and 7 should be replaced with high quality versions.

1- The main contribution of this work is using unlabeled real world road traffic data to mine interpretable specifications improving safety.

2- It would be helpful to provide more details on how the authors come up with the fitness function stated in section 5.2. 

3-The conclusions address the main question posed.

4-The references are accurate and complete.

Reviewer 2 Report

The manuscript submitted for review should be positively evaluated. The authors, with their contribution to science, will confirm the ability to design, conduct research, and draw conclusions. The discussed topic is important and innovative, and the problems raised in it (formulated in the form of research questions) are original. The structure of the work is correct and logical. The authors correctly formulated the research problem and the goals and also defined the research tasks (which is rare). The research could be optionally enriched with hypotheses that will be linked to the goals. Literature and sources have been selected appropriately and to a sufficient degree. The methodology was described in an exemplary manner. In the article, despite my best efforts, I have not found areas that would raise substantive doubts.

Congratulations on your scientific level!

  1. The issues of the article concern the mining/extraction of the structure and parameters of the Sign Temporal Logic (STL) specification from a set of unmarked trajectories, based on grammar. The authors compare the proposed approach with factual (real world) road traffic data, containing several thousand vehicle trajectories. According to the authors, the ability to explore (drill through) the interpreted specifications may contribute to the improvement of traffic safety in the process of its monitoring and planning safety promotion strategies. 

The authors posed two main research questions: 

  1. a) Can we mine specifications that describe the input unlabeled trajectories? 
  2. b) Are the mined specifications readable and interpretable for a human? 

They were formulated correctly. 

  1. The topic is original, actual and timeless. The authors took up a very difficult issue, filling the methodological and empirical gap, because the authors were the first to use unmarked real road traffic data, which increases the cognitive value in this matter. 

  1. As above. This is a deepening of the area of unmarked real data extraction, based on a grammar to identify the structure and parameters of the Sign Temporal Logistic specification, so far not explored to a similar extent. The authors filled the cognitive gap in this respect. 

  1. The article by the authors is an improvement of other approaches already known in the literature. In my personal opinion, the authors should also implement their proposal in the European Union countries in the future and conduct a comparative study showing the results for a different driving culture than in the USA. Such studies are currently lacking. 

  1. The conclusions were formulated correctly, the authors revised the research questions correctly. 

  1. The selection of literature is correct, as mentioned earlier in the review. 

  1. Tables and figures are legible, clear and prepared with due diligence. 

I emphasize that in my opinion the manuscript is a model for other authors and researchers in many dimensions. This is one of the articles that should be submitted to the best article competition. 
